# Exploring Online Activities to Predict the Final Grade of Student

**Silvia Gaftandzhieva [1], Ashis Talukder [2], Nisha Gohain [3], Sadiq Hussain [4,\*], Paraskevi Theodorou [5], Yass Khudheir Salal [6] and Rositsa Doneva [7]**

[1]  Faculty of Mathematics and Informatics, University of Plovdiv "Paisii Hilendarski", 4000 Plovdiv, Bulgaria
[2]  Statistics Discipline, Khulna University, Khulna 9208, Bangladesh
[3]  Directorate of Open and Distance Learning, Dibrugarh University, Assam 786004, India
[4]  Examination Branch, Dibrugarh University, Dibrugarh 786004, India
[5]  Department of Digital Systems, University of Piraeus, 18534 Piraeus, Greece
[6]  Department of Computer Science, South Ural State University, 454080 Chelyabinsk, Russia
[7]  Faculty of Physics and Technology, University of Plovdiv "Paisii Hilendarski", 4000 Plovdiv, Bulgaria
\*  Correspondence: sadiq@dibru.ac.in

**Abstract:** Student success rate is a significant indicator of the quality of the educational services offered at higher education institutions (HEIs). It allows students to make their plans to achieve the set goals and helps teachers to identify the at-risk students and make timely interventions. University decision-makers need reliable data on student success rates to formulate specific and coherent decisions to improve students' academic performance. In recent years, EDM has become an effective tool for exploring data from student activities to predict their final grades. This study presents a case study for predicting the students' final grades based on their activities in Moodle Learning Management System (LMS) and attendance in online lectures conducted via Zoom by applying statistical and machine learning techniques. The data set consists of the final grades for 105 students who study Object-Oriented Programming at the University of Plovdiv during the 2021–2022 year, data for their activities in the online course (7057 records), and attendance to lectures (738). The predictions are based on 46 attributes. The Chi-square test is utilized to assess the association between students' final grades and event context (lectures, source code, exercise, and assignment) and the relationships between attendance at lectures and final results. The logistic regression model is utilized to assess the actual impact of event context on "Fail" students in a multivariate setup. Four machine learning algorithms (Random Forest, XGBoost, KNN, and SVM) are applied using 70% of training data and 30% of test data to predict the students' final grades. Five-fold cross validation was also utilized. The results show correlations between the students' final grades and their activity in the online course and between students' final grades and attendance at lectures. All applied machine learning algorithms performed moderately well predicting the students' final results, as the Random Forest algorithm obtained the highest prediction accuracy—78%. The findings of the study clearly show that the Random Forest algorithm may be used to predict which students will fail after eight weeks. Such data-driven predictions are significant for teachers and decision-makers and allow them to take measures to reduce the number of failed students and identify which types of learning resources or student activities are better predictors of the student's academic performance.

**Keywords:** educational data mining; machine learning; student data; online activity; prediction; final grade

**MSC:** 97P10

## 1. Introduction

In recent years, the digitization of processes in all spheres of public life has led to an increased research interest in extracting knowledge from the collected data for making management decisions to improve the quality of ongoing processes and the achieved results, including in HEIs. Today HEIs use software systems (such as learning management systems, student information systems, human resource systems, research reporting systems, student admission systems, etc.) that store a large amount of data for students and academic and non-academic staff, log their activities, and make the use of technologies for extracting knowledge feasible. The exponential growth of stored data challenges HEIs managers to use data to make data-driven decisions and thus improve the quality of all services offered by HEIs [1]. This comprehensive collection of data will allow HEIs managers to make data-based policies and decisions and form the basis for developing software with artificial intelligence on the ongoing processes in HEIs [2].

Many HEIs receive funding based on the number of students. For this, HEIs managers are looking for ways to reduce dropouts, encourage students to improve their success, and provide a quality education that prepares students well for the labor market so that HEIs be attractive to prospective students. For this reason, student performance is a significant factor for internal and external stakeholder groups in education [3] and a reflection on the quality of the educational services in HEI [4–7]. Student success prediction is a measure of the quality of the teaching offered [8] and determines success at all levels [9]. It allows students to make their plans to achieve the set goals [10], helps teachers adjust learning materials based on the student's ability and identify the at-risk students [11], and helps HEIs managers make better plans to take measures to improve education performance [12]. To improve student performance, many HEIs are increasingly looking for solutions that extract data from information systems, predict final grades before exams and allow HEIs managers/academic staff to make data-driven decisions to encourage students to achieve higher results.

A possible solution that can have a potential impact on supporting HEI managers in making data-driven decisions is Educational Data Mining (EDM) methods. EDM is an interdisciplinary emerging research area that aims to take advantage of the new capabilities of data processing and the maturity of data mining algorithms to enhance the learning process and transform existing information into knowledge [13–15]. EDM utilizes data mining methods in education [16] to improve the education system [17]. EDM analyses educational data (such as student information, education records, exam results, records for participation in online activities and classes, etc.) to develop models for improving learning experiences and institutional effectiveness. Because EDM has to discover knowledge from data stored in multiple sources (such as admissions systems, registration systems, learning management systems, etc.) in different formats and at various granularity levels, each problem needs specific treatment. Traditional data mining techniques cannot handle such issues, and therefore, the knowledge discovery process requires more advanced data mining methods. EDM applies data mining, statistical methods, and machine learning (Decision Trees, Neural Networks, Naïve Bayes, K-Nearest neighbor, etc.) to explore the large-scale data produced by educational organizations to understand the ongoing processes better.

In the last decade, EDM has become extremely valuable for HEIs managers, and there is an increased interest in research. The knowledge acquired from the extracted valuable information from educational data allows HEI managers to improve decision-making processes [18], enhance HEI efficiency [19,20], and thus achieve the highest quality for their students [19]. According to Zhang et al. [21], the advancement of modern management theory and decision-making science and their application in HEIs management will allow HEIs to shift from experience-based management to scientific or information management based on contemporary management theories and methodologies. Therefore, HEIs management will require the existence of a rich toolset to provide the necessary services. Among these tools, we consider that final grade prediction will

play a crucial role in constructing a competitive and effective curriculum benefiting both HEIs managers and students.

Many successful experiments have already been conducted worldwide. Researchers have applied EDM methods to plan courses and predict student enrolment [22], enhance the understanding of the learning process [23], and examine the success chances of curricula. A variety of EDM methods are applied to detect student behavior and preferences [22,24–28], provide feedback and make recommendations to students [22,28–31], and identify student profiles in self-regulated learning [32,33]. EDM methods help teachers identify at-risk students and make corrective strategies to reduce the dropout rate [18,19,22,34–38] and increase students' graduate rates [19,39–42], etc. The final goal of all these studies is to improve student performance. Because of this, a large part of this field's research is devoted to the development of student performance prediction models, which allow for predicting student performance [2,3,8,22,31,43–78].

Lots of studies that predicted final grades in online education motivated us to carry out research in this domain. Meier et al. [79] devised a final grade prediction algorithm for each student in a class. The algorithm explored the student's past history in a course to predict the final grade. Their method yielded 76% prediction accuracy on whether the performance of the students would be poor or not after the 4th week of the course. Their approach confirmed that timely interventions by the instructor were possible based on early in-class assessments. Okubo et al. [80] presented a technique from the log data stored in educational systems by applying a Recurrent Neural Network (RNN) to predict final grades. They utilized the log data of 108 students and examined the prediction accuracy. They compared their RNN technique with multiple regression analysis and confirmed its efficacy in predicting final grades. Xu et al. [81] proposed a grade prediction strategy that exploited student activity features to predict if a learner would clear a test or not to get a certification. Their approach comprised two-step classifications: grade classification and motivation classification. Their proposed technique fitted the classification model at a fine scale. Mouri et al. [82] developed a system for predicting students' final grades and profiled analysis results and visualization based on online logs of 99 first-year students at Kyushu University. The number of logs was approximately 330,000, and their study analyzed and visualized the collected logs. The study's prediction showcased the students who failed to make the grade. Luo et al. [83] examined the impact of students' online behavior patterns on the final grades prediction in blended courses. They designed a predictive model that brought insight into the process of final grades. They clustered the student online behavior into five categories. They demonstrated that the Random Forest classifier showed its efficacy in achieving the highest accuracy.

This study presents a case study for predicting the students' final grades based on their activities in LMS Moodle and attendance in online lectures conducted via Zoom by applying statistical and machine learning techniques. The dataset was balanced by utilizing the single-point crossover method. Different statistical techniques such as chi-square test and regression analysis were applied. Furthermore, machine learning techniques were exploited to find the best-performing classifiers. Experiments were performed on 4-week, 8-week, and full datasets after balancing. The 4-week dataset cannot predict low-performing students, but the 8-week dataset predicts them moderately. Hence, educators can help these low-performing students by engaging them in remedial classes.

We have explored some of the research questions enlisted below and discussed their outcome in the conclusion section:

RQ1. Do the features of learning resources, activities, and attendance of the students demonstrate any correlation with the final academic grade of the learner?

RQ2. Is attendance significantly associated with academic performance?

RQ3. Can machine learning algorithms be utilized to predict the final academic grade of the learner?

The rest of the paper is organized as follows: Section 2 describes the related work, and Section 3 presents the dataset source and the methodology. Section 4 depicts the experiments and results section. Section 5 gives answers to research questions. The last Section concludes with the limitations of the current study and plans for future work in the field.

## 2. Related Work

The basic process of predicting students' performance by using EDM proceeds in several steps: collecting students' historical academic records and labelling them with performance level or GPA, using classification or regression algorithms from machine learning to establish a prediction model, which is trained by the labelled data, and applying the trained model to predict students' performance in various applications after evaluation [12].

The mining of Moodle-based Learning Management Systems (LMS) allows educational policymakers to intercede in the educational process and improve it. It is hard to understand how these machine-learning black-box models operate. To trust the predictions of a model, explainability is the key to making the domain experts understand the reasons behind such predictions. Ljubobratović et al. [69] deployed several tools to explain their model. They explored Moodle activity logs at the University of Rijeka to predict student grades. They exploited Random Forest classifiers for the task. They detected that quizzes and labs were the most vital predictors. Their model showcased an accuracy of 96.3%.

Student performance in exams can be predicted from LMS log data. Hence, predicting students' performance is one of the key factors in achieving the goals of the educational institute. Bhusal [70] employed machine learning tools to predict students' grades on the final examination. Early intervention techniques can aid weak students and positively impact them. The author also predicted the dropout rate of the students. The model demonstrated an accuracy of 0.76.

Moreno-Ger et al. [71] explored different parts of the dataset to understand the most influential factors in predicting students' grades. Their focus was to investigate the best predictors of students' performance in an online university setting. They devised GradeInsight for their specific task. They also utilized Watson Machine Learning services to derive the vital predictors in the student's final grades and incorporate them into the instructor's support systems.

Early-warning systems based on Moodle logs can hint at the course administrators about the student's achievements in the examinations. Quinn et al. [72] introduced a model to assess the student's academic performance in a blended educational setting. They predicted whether the student would pass or fail, as well as the letter grade of the student. Their model achieved an accuracy of 92.2% in predicting whether the student would pass or fail and 60.5% in predicting the student's academic grade. The first ten weeks of Moodle logs could predict the failing students with higher accuracy, while the first six weeks of data failed to do that precisely.

Mueen et al. [73] utilized data mining approaches to mine the students' academic performance. Their model could assist the instructors in locating the poor-performing students and providing more attention to improve their performance. They employed Naïve Bayes, Multilayer Perception, and Decision tree (C4.5) classifiers on the undergraduate student's data. Naïve Bayes proved its efficacy by yielding an accuracy of 86% and outperforming the other two classifiers. They suggested assigning a fair number of marks to the messages posted on the forum.

Gadhavi et al. [74] devised a univariate linear regression model to know the performance of the final examination in advance. They considered the internal assessment marks to predict the final grade in that particular subject. To get accurate predictions, they normalized the internal assessment mark to 100. Their model helped the students

know in advance how many marks are needed in internal examinations to achieve a specific grade.

Various factors that affect the student's achievement and learning process make students' academic performance prediction problems complex issues. The online activity data and assessment grades were investigated by Alhassan et al. [75] to measure their impact on the student's overall performance. They employed logistic regression, multi-layer perceptron, sequential minimal optimization, random forest, and decision tree classifiers on the LMS data for the prediction of student academic performance. Random forest followed by decision tree algorithms outperformed their counterparts in such predictions. In this regard, both the activity data and assessment grades demonstrated better performance than considering activity data alone.

Educational data mining tools may explore the hidden relationship between students' academic performance and educational data. The author [2] utilized the mid-term grades of the students to develop a machine learning model for the prediction of the final examination grades of the students. The dataset consisted of 1854 Turkish students' academic records for the Turkish Language-I course from 2019–2020. He applied k-nearest neighbor, Naive Bayes, logistic regression, support vector machines, nearest neighbor, and Random Forests classification techniques. Their model yielded 70–75% classification accuracy. The parameters the author used were Faculty Data, Department Data, and mid-term exam grades.

Qiu et al. [76] presented the behavior classification-based e-learning performance (BCEP) model for predicting learning performance by applying machine learning. They also designed the process-behavior classification (PBC) framework to assess online behavior based on an e-learning environment. They used feature fusion with behavioral data and established empirically that their BCEP approach had a decent prediction effect. The performance of the PBC model was also superior to the traditional classification techniques.

Mozahem [77] exploited the students' data from a private university in Lebanon that utilized LMS to augment face-to-face teaching. Data from eight courses were collected across two semesters. He examined whether the grade of the students and gender had any association with logging into the system by analyzing the event history. His results confirmed that login activity had a positive impact on the academic performance of the students. Female students logged in to the system more frequently than their male counterparts.

Different machine learning clustering and classification methods were employed by Hussain et al. [78] to predict poor-performing students from Moodle data in a massive open online course (MOOC) environment. If the instructors know such low-performing students in advance, then such students may be facilitated with some remedial sessions. They exploited the fuzzy unordered rule induction algorithm (FURIA) classification method for different group categories of students during the Moodle course. An effective educational environment could be created by sending alerts to low-performing and inactive students. They suggested incorporating their model into the Moodle system to achieve educational goals.

Online education has been a center of attraction for educational data mining researchers for both small private online courses (SPOCs) and massive open online courses (MOOCs) [84]. Although there are many studies related to online education, the knowledge gap includes student performance, engagement, and retention assessments. With the increased challenges and demands in online education, different researchers exploited machine learning and deep learning strategies to tackle the issues such as performance and dropout in student outcomes. Prevailing prediction techniques are still inadequate to identify the most suitable methods for the prediction outcome. There is a need for specialized techniques to explore the big data repositories of student records and extract the needed information from them. Moreover, different HEIs across the globe have diverged datasets with different features making it more complex to satisfy their

needs from it with a single system. This research gap inspires the researchers to apply various methods so that the stakeholders could potentially benefit from them.

A clickstream or log data is the initial form of learning-behavior data [85]. The events such as submitting an assignment are the major variables in assessing educational outcomes. Learning behavior patterns may be categorized into four forms of features: raw, raw-temporal, statistical, and statistical-temporal features. Most studies utilize a coarse-grained statistical approach to represent learning events in terms of frequency, length, rate, and accumulation over a specific time frame. Deep learning algorithms are employed these days to derive temporal properties from raw data. Deep learning models with convolution layers have been applied to automatically extract the most salient features for the prediction task. Statistical features are the most commonly used feature engineering strategy in both performance prediction and dropout prediction tasks. In contrast, raw-temporal features were applied in a small portion of studies. In the review by [85], they observed that the most commonly utilized features in the predictive model for academic grading are demographics, academic background, interaction, assessment, enrolment, course, video interaction, etc. The final grades, cumulative grade point average (CGPA), and internal assessment are applied as the key features in several studies related to online education students' performance [7].

## 3. Materials and Methods

### 3.1. Dataset

The data set includes data on the performance of 105 students in the course "Object Oriented Programming", the final grades obtained, and the attendance of the classes. During the COVID-19 pandemic, students are learning online using Moodle and Zoom. The online course "Object-oriented programming" is published on Moodle. The course consists of eight modules. Each module includes a lecture with theoretical material, programming codes of the examples presented in the lectures, tasks for exercises with their solution, and tasks for self-assessment. Students should read all the theoretical knowledge presented in the module and test programming codes. In addition, students can try to modify some of their functionalities. After testing solutions to all tasks for exercise, students have the opportunity to test their knowledge by trying to solve the self-assessment tasks. Each week, students have an online lecture and exercise conducted via Zoom.

To comply with the European Union General Data Protection Regulation, we removed the first five digits from students' faculty numbers that store information about the year, the faculty identification number, and the study program. In this way, we anonymized students. Data for student performance and attendance are retrieved from Moodle, where the online course is published, and the Zoom video conferencing platform, which is used to deliver the online classes.

The dataset extracted from Moodle includes only the data for student activity in the course—Lecture (View), Source code (View), Exercise (View), and Assignment (Uploaded/Submitted). Since only viewing the assignment does not prove an attempt to solve the specific task, all records in which the action is "View" are excluded from the data set. The final dataset with data from the online course in Moodle consists of 7057 records, part of which are presented in Figure 1.

To form the dataset, data on the student's attendance in the online classes, which were held every week, were also extracted. Figure 2 presents some of the data extracted from Zoom. The total number of records extracted from Zoom is 738.

The final merged dataset used for this study is comprised of 105 student records from Plovdiv University with 46 attributes (see Figure 3), as shown in Table 1. For brevity, all attributes that refer to activities of the same type are presented in 1 row. Therefore, for example, row 2 presents eight attributes from the dataset, one for each lecture.

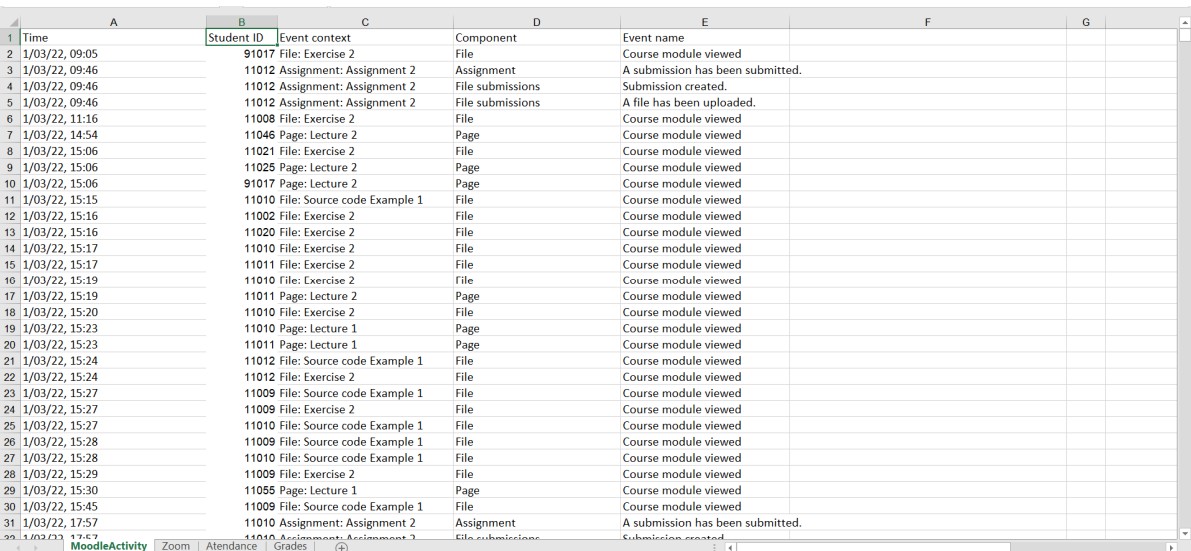

**Figure 1.** Data from Moodle.

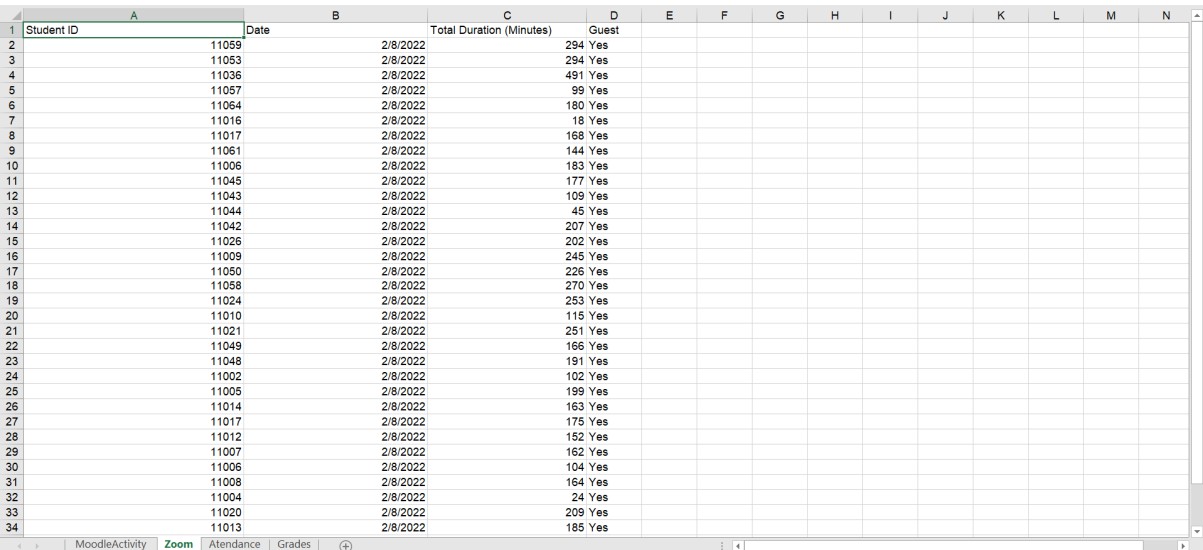

**Figure 2.** Data from Zoom.

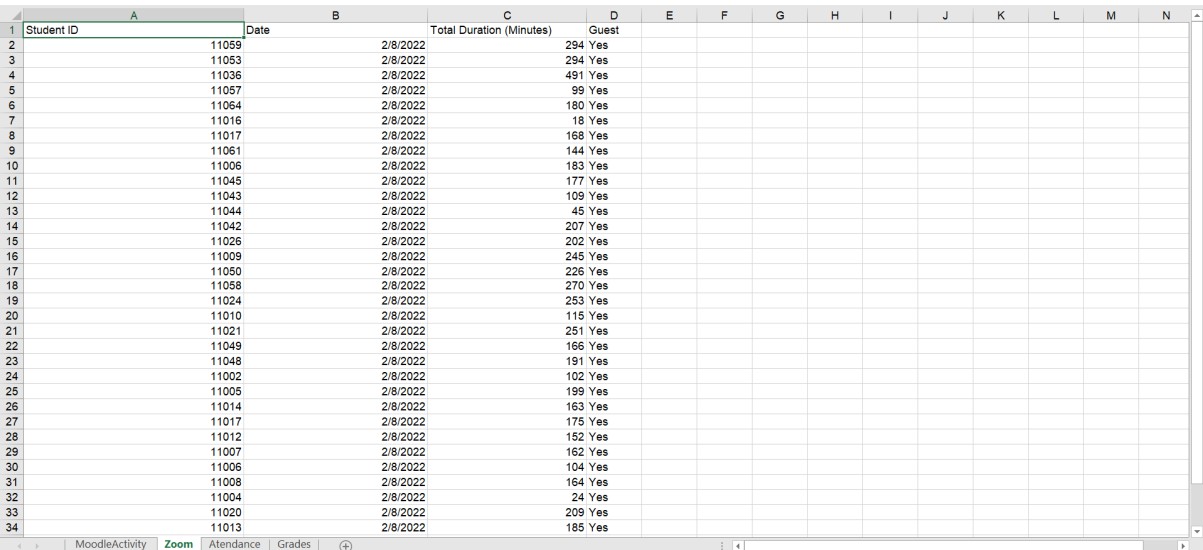

**Figure 3.** Final dataset.

**Table 1.** Attribute Description.

| Attribute/s | Description |
| --- | --- |
| GRADE | Final semester mark obtained in the "Object-oriented programming" course—Fail (2), Satisfactory (3), Good (4), Very Good (5), and Excellent (6) |
| LEC1-LECT8 | Eight attributes indicating whether the student has read each lecture or not (Lecture 1–Lecture 8) |
| ASSIGN1-ASSIGN8 | Eight attributes indicating whether the student has submitted a task for self-assessment (Assignment 1–Assignment 8) |
| CODE1-CODE8 | Eight attributes indicating whether the student has read each programming code or not (Source code 1–Source code 8) |
| EXER1-EXER8 | Eight attributes indicating whether the student has read each exercise or not (Exercise 1–Exercise 8) |
| ATTEND1-ATTEND13 | Eight attributes indicating whether the student has attended the online classes or not (Week 1–Week 13) |

The training set consists of the final grades for 105 students, 7057 records for their activities in the online course, and 738 records for their attendance in online lectures.

*3.2. Methodology*

We applied the chi-square test to assess the association between academic performance (grade) and event context (lectures, source code, exercise, and assignment). *p*-values under 0.05 were regarded as significant. All machine learning (ML) models incorporated the following factors (Event context) relevant to academic performance: lectures, exercises, source codes, and assignments. To conduct data analyses, Python software and SPSS version 23 were used to conduct data analyses.

Afterward, four machine learning (ML) methods (Random Forest, Extreme Gradient Boosting, K-nearest neighbors, and Support Vector Machine) were applied using a sample of 70% of the participants in each group (training dataset) and verified in the remaining 30% of the data (test dataset). Moreover, the k-fold cross validation technique was also utilized. This is a useful tool when the size of the dataset is relatively small, as in our investigation [86]. Using the k-fold cross validation method, the dataset is divided into subgroups of the necessary sizes at random. In order to develop the model, one subset was randomly used for the testing set, and the remaining subsets were utilized for the training set. The final result of the model is generated from the average result of the testing set after this process has been performed K times.

We evaluated the performances of the ML-classifiers using accuracy, precision, recall, and F1-Score. The following equations are used to calculate the evaluation matrices [87]:

$$\text{Accuracy} = \frac{\text{TP} + \text{TN}}{\text{TP} + \text{FP} + \text{FN} + \text{TN}}$$

$$\text{Precision} = \frac{\text{TP}}{\text{FP} + \text{TP}}$$

$$\text{Recall} = \frac{\text{TP}}{\text{FN} + \text{TP}}$$

$$\text{F}_1 = 2 \times \frac{\text{Precision} \times \text{Recall}}{\text{Precision} + \text{Recall}}$$

where TP is true positive rate, TN is true negative, FP is false positive rate, and FN is false negative rate.

Random Forest (RF): For better classification and regression results, more than one tree is ideal. In order to produce more accurate results, Random Forest is constructed using many decision trees. In order to combine the results, it first generates bootstraps by randomly resampling data from the training dataset. In the RF approach, training is modeled using bootstrap aggregation. By averaging the projected values of each tree, the trained model may predict an unknown sample.

Extreme Gradient Boosting (XGBoost): Gradient Boosted decision trees are implemented using XGBoost technology. Decision trees are generated sequentially in this approach. All independent variables are given weights, which are subsequently used to

feed information into the decision tree that forecasts outcomes. Variables that the tree incorrectly predicted are given more weight before being placed into the second decision tree. These distinct classifiers/predictors are then combined to produce a robust and accurate model. It applies to user-defined prediction issues as well as regression, classification, and ranking issues.

k-nearest neighbors (KNN): The supervised ML family of algorithms includes the robust and adaptable classifier known as k-nearest neighbors (KNN). Due to the fact that it makes no explicit assumptions regarding the distribution of the dataset, KNN is a non-parametric algorithm. This method sorts newly discovered instances based on a similarity metric and records every single accessible case. A case is assigned to a class based on a majority of the votes cast by its neighbors, with the case's k-closest neighbors being determined via a distance function, and the case is placed in the class that is generally regular among those neighbors [88].

Support Vector Machine (SVM): An effective supervised machine learning (ML) technique for classification and regression issues is the Support Vector Machine (SVM). Through a hyperplane, it is a mechanism that is most effective in separating the two classes. It operates on the presumption that only the support vectors are important, ignoring other training samples. High-dimensional spaces can use this classifier successfully [89]. Additionally, the experiments employed the radial basis function (RBF) kernel.

Single-point Crossover: For balancing the dataset, we apply single-point crossover. Crossover refers to the random recombination of individuals that results in children with different genetic makeups (exchange of chromosomal coding sequence segments) [90]. The crossover probability, which is the fundamental variable in the crossover, indicates the likelihood that the crossover will occur. Single-point, multi-point, heuristic, and mathematical crossovers are common crossover techniques.

A single point, also known as the pivot point or crossover point, is used in the single-point crossover procedure to cut the chosen parent population or the two mating chromosomes. To create two offspring chromosomes, the genetic material to the left (or right) of the point is switched between the two parent chromosomes at this point (cut).

## 4. Results

### 4.1. Univariate Analysis

Figure 4 displays the percentage of students according to grade level. The bar graph shows that 32.4% of the students whose academic achievement is "Good" have the greatest percentage. Additionally, a little more than 21% of students' academic achievement falls into the "Satisfactory" group. However, the students whose academic achievement is "Excellent" are given the lowest percentage (9.5%). The percentages for "Fail" and "Very Good" academic performance categories are nearly the same in the above bar graph. Academic performance for 18.1% of students is classified as "Fail", while for 19% of students it is "Very Good."

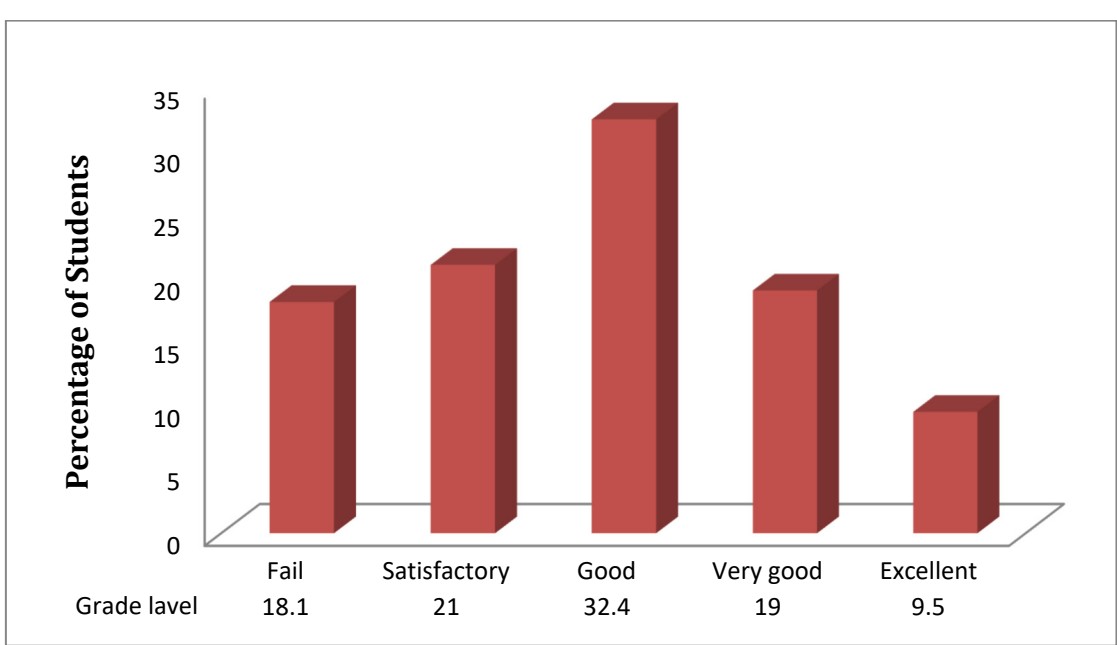

**Figure 4.** Percentage of students based on grade level.

It is clearly observed that the dataset is imbalanced. The dataset was balanced by applying a single-point crossover technique. Now, according to grade level, Figure 5 displays the percentage of students. The pie chart (Figure 5) shows that 18% of the students whose academic achievement is "Good", have the lowest percentage. Additionally, a little more than 21% of students' academic achievement falls into the "Excellent" and "Very good" groups. Academic performance for 20.1% of students is classified as "Fail", while for 19.6% of students it is "Satisfactory". The percentages for all groups of academic performance categories are nearly the same in the above pie chart, indicating that the data are balanced.

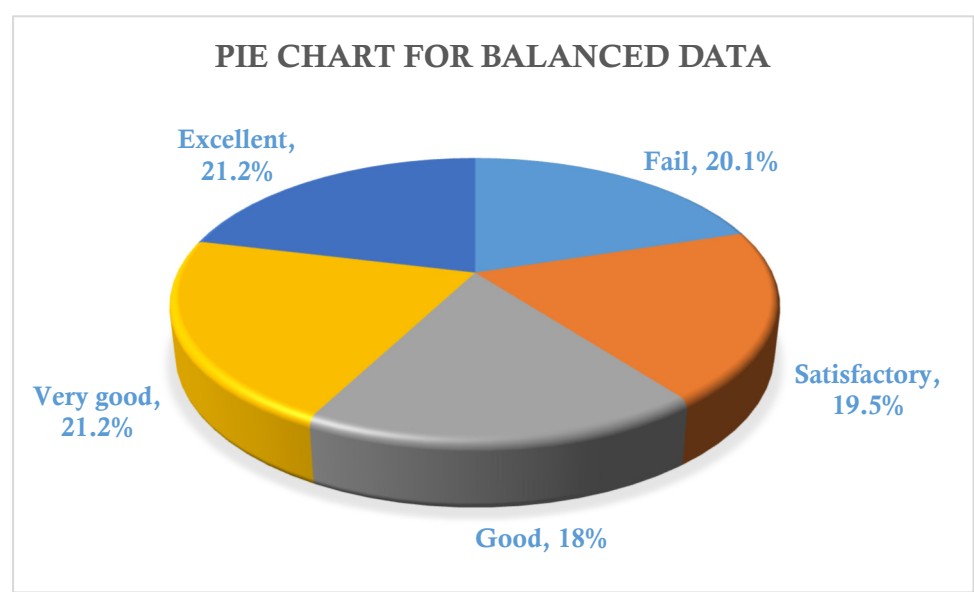

**Figure 5.** Percentage of students based on grade level (balanced data).

Figure 6 describes the relationship between the students who had submitted the Assignment 2 and the grade they attained. The submission of this assignment is of utmost importance to the successful completion of the course, as it requires the demonstration of basic knowledge of the discipline, namely, class design and declaration. From

the figure, it is clear that most of the students who had submitted Assignment 2 were placed in Grade 4 (i.e., Good).

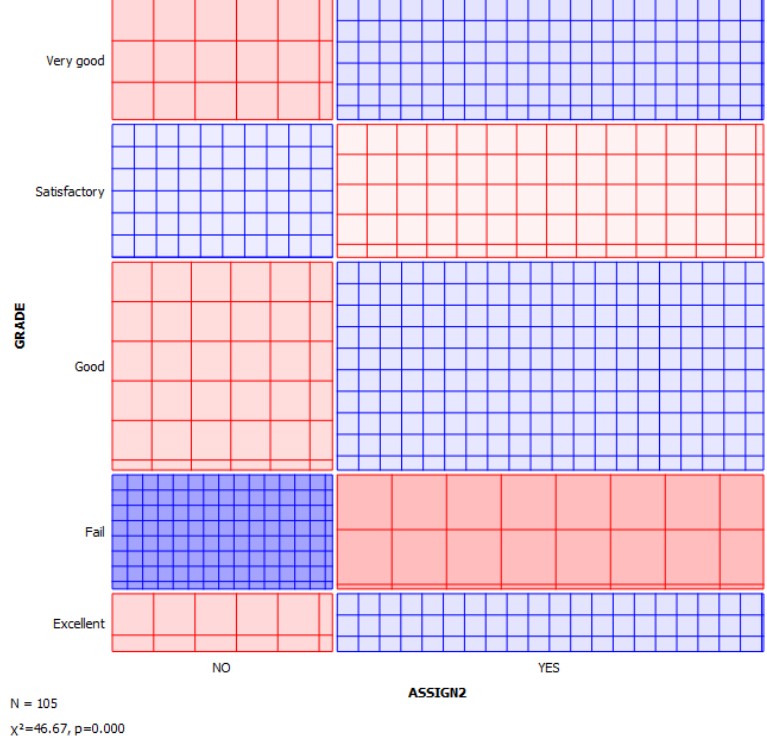

**Figure 6.** Sieve Diagram.

Figure 7 depicts the relationship between the students' grade and whether he has completed (course module viewed) two of the key student activities for successful completion of the course—Lecture 7 and Exercise 8. It is observed that majority of students who had not viewed Lecture 7 and Exercise 8 were among "Fail" grade students.

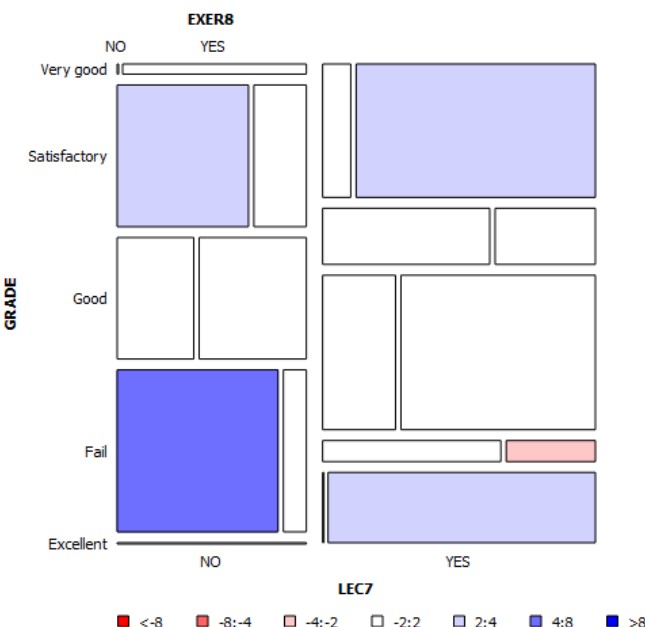

**Figure 7.** Mosaic display.

*4.2. Bivariate Analysis*

Table 2 shows the bivariate relationship between academic performance (grade), one type of activity (assignments), and learning material (lectures, exercises, and source codes). Assignments are one sort of activity among lectures, exercises, and source codes that make up the learning content. Academic performance is significantly correlated with all event contexts. Compared to the other categories of academic performance, most students who are using Lecture 1 as their learning context were able to obtain "Good" academic performance. However, Lecture 6 has the greatest rate (36.8%) of students that achieve "Good" academic achievement. Additionally, Lecture 7 contributes the highest percentages with 30.6% and 16.1%, respectively, for "very excellent" and "Excellent" academic performance.

Moving on to the "Exercises" learning context, all exercise types demonstrated a substantial correlation with academic achievement. There is zero percent failure among the students who do Exercise 3. The highest percentage (21%) for a satisfactory result is discovered for Exercise 2. Of all the activities, Exercise 6 aids students in achieving the highest percentage of "Good" academic performance (41.2%). The Exercises 5 and 8 have the greatest percentages (16.4%), so students who want to succeed academically at an "Excellent" level should use those as their learning contexts.

One of the learning environments is source code. There are nine different sorts of source codes, and each one is significantly correlated with academic achievement. Source code example 3 displays the lowest percentage (0%), and Source code example 1 demonstrates the highest percentage when "Fail" is selected (7.7%). With 22.8% and 28.1%, respectively, Source code example 3 has the highest percentage of both "Satisfactory" and "Good" academic performance. The majority of students (around 20%) use Source code example 8 as an environment for learning in order to produce "Excellent" academic results. Similarly, the learning context Source code example 9 performs best when "Good" academic performance is considered, as it contains the largest percentage (41%).

With the largest percentages of "Satisfactory" and "Good" academic performance (17.4% and 40.6%, respectively), Assignment 2 is more practical for the students, according to Table 1. However, due to its higher percentages, Assignment 7 is more effective for achieving "very good" academic performance, whereas Assignment 5 is for "Excellent" academic performance.

**Table 2.** Assessing the association between learning content (lectures, exercises, and source codes), one type of activity (assignment), and academic performance (Grade) with *p*-values obtained from a chi-square test.

| Event Context | GRADE | | | | | *p* Value |
| | Fail *n*(%) | Satisfactory *n*(%) | Good *n*(%) | Very good *n*(%) | Excellent *n*(%) | |
| --- | --- | --- | --- | --- | --- | --- |
| | Lectures | | | | | |
| LEC1 | 7 (8.5) | 18 (22.0) | 30 (36.6) | 18 (22.0) | 9 (11.0) | <0.001 |
| LEC2 | 10 (11.0) | 20 (22.0) | 31 (34.1) | 20 (22.0) | 10 (11.0) | <0.001 |
| LEC3 | 8 (9.6) | 17 (20.5) | 29 (34.9) | 19 (22.9) | 10 (12.0) | <0.001 |
| LEC4 | 5 (6.6) | 13 (17.1) | 28 (36.8) | 20 (26.3) | 10 (13.2) | <0.001 |
| LEC5 | 6 (8.0) | 13 (17.3) | 26 (34.7) | 20 (26.7) | 10 (13.3) | <0.001 |
| LEC6 | 6 (8.2) | 10 (13.7) | 28 (38.4) | 19 (26.0) | 10 (13.7) | <0.001 |
| LEC7 | 3 (4.8) | 8 (12.9) | 22 (35.5) | 19 (30.6) | 10 (16.1) | <0.001 |
| LEC8 | 3 (4.6) | 9 (13.8) | 24 (36.9) | 19 (29.2) | 10 (15.4) | <0.001 |
| | Exercises | | | | | |
| EXER2 | 4 (5.1) | 17 (21.5) | 29 (36.7) | 19 (24.1) | 10 (12.7) | <0.001 |
| EXER3 | 0 (0.0) | 14 (20.6) | 27 (39.7) | 17 (25.0) | 10 (14.7) | <0.001 |

| | | | | | |
|---|---|---|---|---|---|
| EXER4 | 3 (4.3) | 11 (15.9) | 25 (36.2) | 20 (29.0) | 10 (14.5) | <0.001 |
| EXER5 | 2 (3.3) | 9 (14.8) | 21 (34.4) | 19 (31.1) | 10 (16.4) | <0.001 |
| EXER6 | 2 (5.9) | 6 (17.6) | 14 (41.2) | 9 (26.5) | 3 (8.8) | <0.001 |
| EXER7 | 4 (6.1) | 10 (15.2) | 23 (34.8) | 19 (28.8) | 10 (15.2) | <0.001 |
| EXER8 | 3 (4.9) | 7 (11.5) | 23 (37.7) | 18 (29.5) | 10 (16.4) | <0.001 |
| **Source codes** | | | | | |
| SC1 | 6 (7.7) | 17 (21.8) | 26 (33.3) | 19 (24.4) | 10 (12.8) | <0.001 |
| SC2 | 3 (5.1) | 13 (22.0) | 19 (32.2) | 15 (25.4) | 9 (15.3) | <0.001 |
| SC3 | 0 (0.0) | 13 (22.8) | 18 (31.6) | 16 (28.1) | 10 (17.5) | <0.001 |
| SC4 | 1 (2.0) | 9 (18.4) | 19 (38.8) | 12 (24.5) | 8 (16.3) | <0.001 |
| SC5 | 2 (4.8) | 6 (14.3) | 17 (40.5) | 9 (21.4) | 8 (19.0) | <0.001 |
| SC6 | 3 (5.5) | 11 (20.0) | 21 (38.2) | 11 (20.0) | 9 (16.4) | <0.001 |
| SC7 | 1 (2.8) | 6 (16.7) | 14 (38.9) | 9 (25.0) | 6 (16.7) | <0.001 |
| SC8 | 2 (5.7) | 5 (14.3) | 12 (34.3) | 9 (25.7) | 7 (20.0) | <0.001 |
| SC9 | 2 (5.1%) | 6 (15.4) | 16 (41.0) | 8 (20.5) | 7 (17.9) | <0.001 |
| **Assignments** | | | | | |
| ASSIG2 | 1 (1.4) | 12 (17.4) | 28 (40.6) | 18 (26.1) | 10 (14.5) | <0.001 |
| ASSIG3 | 0 (0.0) | 7 (13.5) | 21 (40.4) | 14 (26.9) | 10 (19.2) | <0.001 |
| ASSIG4 | 2 (3.5) | 7 (12.3) | 21 (36.8) | 17 (29.8) | 10 (17.5) | <0.001 |
| ASSIG5 | 1 (2.6) | 3 (7.7) | 11 (28.2) | 14 (35.9) | 10 (25.6) | <0.001 |
| ASSIG6 | 2 (5.1) | 2 (7.7) | 11 (28.2) | 15 (38.5) | 8 (20.5) | <0.001 |
| ASSIG7 | 1 (3.0) | 1 (3.0) | 10 (30.3) | 15 (45.5) | 6 (18.2) | <0.001 |
| ASSIG8 | 1 (2.7) | 3 (8.1) | 14 (37.8) | 11 (29.7) | 8 (21.6) | <0.001 |

Note: LEC = Lecture; EXER = Exercise; SC = Source code; ASSIGN: Assignment.

Table 3 represents the association between academic performance (Grade) and attendance of the students. Student's attendance is significantly associated with academic performance. Most students who had more than 60% attendance achieved Good (37.7%), Very Good (32.1%), and Excellent (18.9%) academic grades when compared to the other categories of academic achievement.

**Table 3.** Evaluating the relationship between attendance and grade using *p*-values from the chi-square test.

| | **GRADE** | | | | | |
|---|---|---|---|---|---|---|
| **Status** | **Fail** *n*(%) | **Satisfactory** *n*(%) | **Good** *n*(%) | **Very good** *n*(%) | **Excellent** *n*(%) | ***p* Value** |
| **Attendance** | | | | | | |
| Less than 60% | 16 (30.8) | 19 (36.5) | 14 (26.9) | 3 (5.8) | 0 (0.0) | <0.001 |
| 60% or more | 3 (5.7) | 3 (5.7) | 20 (37.7) | 17 (32.1) | 10 (18.9) | |

*4.3. Regression Analysis*

We used a logistic regression model and divided our main response variable (Grade) into two categories: Fail and not fail, to determine the actual influence of the "Event context" on "Fail" students. The regression results were shown in Table 4. We found that the majority of lectures significantly affected students' final grades. For instance, the odds of failing are lower for students who attended LEC1 (OR = 0.079; *p* < 0.001). The ultimate grade a student receives is significantly impacted by LEC2, LEC3, LEC4, LEC5, and LEC8. It is clear from looking at various Event contexts that EXER2, EXER5, EXER7, EXER8, SC 1, SC4, ASSIG2, ASSIG4, and ASSIG8 have a major impact on a student's ultimate grade.

**Table 4.** Maximum likelihood estimates of logistic regression model.

| Event Context | Estimates | Odds Ratio (OR) | $p$ Value |
|---|---|---|---|
| Lectures | | | |
| LEC1 | −2.539 | 0.079 | <0.001 |
| LEC2 | −3.080 | 0.046 | <0.001 |
| LEC3 | −2.523 | 0.080 | <0.001 |
| LEC4 | −2.746 | 0.064 | <0.001 |
| LEC5 | −2.439 | 0.087 | <0.001 |
| LEC6 | 0.254 | 1.289 | 0.663 |
| LEC7 | −1.044 | 0.352 | 0.218 |
| LEC8 | −1.851 | 0.157 | 0.017 |
| Exercises | | | |
| EXER2 | −2.325 | 0.098 | <0.001 |
| EXER3 | 21.545 | 0.000 | 0.995 |
| EXER4 | −0.815 | 0.442 | 0.224 |
| EXER5 | −1.976 | 0.139 | 0.005 |
| EXER6 | 0.528 | 1.696 | 0.408 |
| EXER7 | 1.444 | 4.238 | 0.017 |
| EXER8 | −1.451 | 0.234 | 0.029 |
| Source codes | | | |
| SC1 | −1.965 | 0.140 | <0.001 |
| SC2 | −0.944 | 0.389 | 0.106 |
| SC3 | −21.177 | 0.000 | 0.996 |
| SC4 | −2.494 | 0.083 | 0.002 |
| SC5 | −1.224 | 0.294 | 0.064 |
| SC6 | −1.525 | 0.218 | 0.062 |
| SC7 | −1.949 | 0.142 | 0.058 |
| SC8 | 1.601 | 4.956 | 0.188 |
| SC9 | −0.389 | 0.677 | 0.699 |
| Assignments | | | |
| ASSIG2 | −3.708 | 0.025 | <0.001 |
| ASSIG3 | −18.260 | 0.000 | 0.996 |
| ASSIG4 | −1.862 | 0.155 | 0.016 |
| ASSIG5 | 0.080 | 1.084 | 0.940 |
| ASSIG6 | −0.553 | 0.575 | 0.429 |
| ASSIG7 | −1.665 | 0.189 | 0.054 |
| ASSIG8 | −2.101 | 0.122 | <0.01 |

Table 5 depicts the Accuracy, Precision, Recall, and F1-score of the best classifier RF fold-wise for five-fold cross-validation. If we explore the results group-wise on the test dataset, the "Excellent" group performed best among other groups followed by the "Fail" and "Very Good" groups.

**Table 5.** Fold-wise evaluation metrics of Random Forest Classifier.

| Random Forest Algorithm | Accuracy | Precision | Recall | F1-Score |
|---|---|---|---|---|
| K = 0 | 0.91 | 0.90 | 0.90 | 0.90 |
| K = 1 | 0.77 | 0.72 | 0.72 | 0.72 |
| K = 2 | 0.77 | 0.57 | 0.40 | 0.47 |
| K = 3 | 0.69 | 0.98 | 0.90 | 0.94 |
| K = 4 | 0.76 | 0.71 | 0.98 | 0.83 |
| Average | 0.78 | 0.77 | 0.78 | 0.77 |

*4.4. Machine Learning Algorithms*

Data pre-preparing techniques, such as single-point crossover, are used before machine learning (ML) algorithms are employed. We also consider four weeks' and eight weeks' dataset to compare the performance measures of the ML models using 70% data for training and rest 30% for testing. All of the algorithms performed with a moderate accuracy score (>60%), as per the accuracy matrices shown in Table 6. The maximum accuracy for a full dataset was reached by Random Forest at 71%, followed by XGBoost, KNN, and SVM at 68%, 65%, and 66%, respectively. The precision value for each of the four ML models was greater than 70%. The highest recall value was attained by Random Forest (78%), followed by XGBoost (76%), SVM (73%), and KNN (70%). Random Forest received the greatest F1-score of 77%, followed by XGBoost, and KNN with scores of 75 and 73%, respectively. SVM, on the other hand, received a 70%. The most intriguing result is that our machine learning (ML) algorithms did not perform well when applied to a dataset with a four-week time frame but improved and nearly met expectations after eight weeks. As a result, we will be able to identify the students who will fail after eight weeks.

**Table 6.** Comparison of ML classification algorithms based on different performance indicators by utilizing full data, 4-weeks data and 8-weeks data (using 70% training set and 30% testing set).

| Algorithms | Four (4) Weeks Data | Eight (8) Weeks Data | Imbalanced Complete Dataset | Balanced Complete Dataset | Four (4) Weeks Data | Eight (8) Weeks Data | Imbalanced Complete Dataset | Balanced Complete Dataset | Four (4) Weeks Data | Eight (8) Weeks Data | Imbalanced Complete Dataset | Balanced Complete Dataset | Four (4) Weeks Data | Eight (8) Weeks Data | Imbalanced Complete Dataset | Balanced Complete Dataset |
|---|---|---|---|---|---|---|---|---|---|---|---|---|---|---|---|---|
| | Accuracy | Accuracy | Accuracy | Accuracy | Precision | Precision | Precision | Precision | Recall | Recall | Recall | Recall | F1-Score | F1-Score | F1-Score | F1-Score |
| RF | 0.47 | 0.71 | 0.70 | 0.78 | 0.51 | 0.75 | 0.77 | 0.77 | 0.53 | 0.70 | 0.72 | 0.78 | 0.46 | 0.71 | 0.71 | 0.77 |
| XGB | 0.43 | 0.68 | 0.59 | 0.76 | 0.56 | 0.68 | 0.56 | 0.76 | 0.59 | 0.68 | 0.59 | 0.76 | 0.55 | 0.67 | 0.57 | 0.75 |
| KNN | 0.44 | 0.65 | 0.63 | 0.72 | 0.48 | 0.66 | 0.63 | 0.73 | 0.45 | 0.69 | 0.66 | 0.73 | 0.54 | 0.62 | 0.60 | 0.73 |
| SVM | 0.39 | 0.66 | 0.59 | 0.70 | 0.40 | 0.68 | 0.89 | 0.72 | 0.49 | 0.67 | 0.59 | 0.70 | 0.45 | 0.67 | 0.55 | 0.70 |

Note: RF = Random Foreest; XGB = XGBoost; KNN = K nearest neighbour; SVM = Support Vector Machine.

## 5. Discussion

The presented results allow to give answers on the research questions.

RQ1. Do the features of learning resources, activities, and attendance of the students demonstrate any correlation with the final academic grade of the learner?

Specifically, we considered three types of learning resources: lectures, exercises, and source codes; one type of activity, i.e., assignment, and the attendance of the students to the lectures, as the features for our study. Our analysis showed that students' final grades are significantly correlated with the student's participation in different types of activities and learning resources. We have also established some correlations between different types of lectures, exercises, and source codes and found that the students' academic performance is substantially correlated with some specific types of lectures, exercises, and assignments—for example, students participating in Lecture 7 exhibit "Excellent" academic rating. The logistic regression model also confirmed the actual impact of event

context on "Fail" students in a multivariate group. EXER2, EXER5, EXER7, EXER8, SC 1, SC4, ASSIG2, ASSIG4, and ASSIG8 majorly impact a student's ultimate grade.

RQ2. Is attendance significantly associated with academic performance?

Academic performance is strongly correlated with student attendance. When compared to the other categories of academic accomplishment, the majority of students who had more than 60% attendance earned Good (37.7%), Very Good (32.1%), and Excellent (18.9%) academic ratings. These results are in line with findings from other related works, further solidifying the observations. Aydoğdu [91] argued empirically that the variables attendance number to live classes, attendance number to archived courses, and time spent on the content contributed most to the prediction of the output variable. Wojciechowski et al. [92] confirmed that by applying regression analysis, two variables served as the best predictors: attendance at an orientation session and the student's grade point average. The positive correlations between attendance rate and learning outcomes had also been supported by Jo et al. [93].

RQ3. Can machine learning algorithms be utilized to predict the final academic grade of the learner?

We tried to answer this question by training predictive classifiers for the student's academic performance based on the correlated features. In our study, we used four machine learning algorithms—Random Forest, XGBoost, KNN, and SVM—to predict students' performance based on the chosen features. According to our findings, student's academic achievement was predicted by all four machine learning algorithms just fair to slightly well. With a prediction accuracy of 78%, the Random Forest method performed better than the other three algorithms. With a recall value of 72%, Random Forest outperformed SVM in terms of performance. Random Forest exhibited the highest F1-score of 0.77, while SVM demonstrated the lowest of it at 0.70. For imbalance dataset, the accuracy was found to be 0.70 whereas after balancing and using 70% and 30% data for both training and testing and applying 5-fold cross validation, the accuracy was improved to 0.78. In the testing dataset for classification by RF, the "Excellent" group exhibited superior performance compared to other groups while the outcome of the "Satisfactory" group was found inferior. As the study is a retrospective one, actual intervention was not possible. However, if we explore the four weeks' dataset and eight weeks' dataset, it is possible to predict the low-performing students after 8 weeks. This will be of great help for educators and policymakers to pay extra attention to enhancing their grades. Hence, for future online courses, after a specific period, i.e., 8 weeks for this particular program, the educators may find poorly performing students with the help of our model. Thus, it is possible to use students' actions on the Learning Management System and attendance in online lectures to predict their academic performance in the course reasonably well with the help of machine learning algorithms.

We compare our study with some of the existing studies in this domain in Table 7. As the datasets applied in these studies are different; hence, a fair comparison is not possible and also, one best classifier cannot be derived as classifiers showcased its efficacy differently as the dataset changed.

**Table 7.** Comparison of our study with some of the existing works in the domain.

| Reference | Dataset | Courses | Records | Features | Algorithms Used | Performance Matrices |
|---|---|---|---|---|---|---|
| [94] | Open University Learning Analytics Dataset (OULAD) | 22 | 32,593 | Demographic, Registration, Assessment, Interaction | J48, RepTree, RandomTree, FURIA | Accuracy 92.56%(for best classifier FURIA) |
| [95] | OULAD | 22 | 32,593 | Demographic, Registration, Assessment, Interaction | LightGBM, XGBoost, AdaBoost, RF, MLP and Naïve Bayes | Precision, Recall, F1-score 84.1% (for best classifier LightGBM) |

| Ref | Dataset | Courses | Records | Attributes | Algorithms | Evaluation |
|---|---|---|---|---|---|---|
| [96] | OULAD | 22 | 32,593 | Demographic, Registration, Assessment, Interaction | Long-Short Term Memory (LSTM), SVM, Logistic Regression, DOPPFCN | Precision, Recall, F1-score 0.8757 F1-score for intra-class binary classification |
| [69] | University of Rijeka Dataset | One | 408 | ID, lectures, quizzes, labs, videos and grade | Random Forest | Accuracy 96.3% |
| [72] | Further education setting Dataset | One | 690 | Moodle log data variables | Random Forest, LDA, k-NN, GBM | Accuracy, Kappa, Accuracy 60.5% (for best classifier Random Forest) |
| [73] | King Abdulaziz University Dataset | Two | 60 | assignments, quizzes, forums and tests | Naïve Bayes, Neural Network, and Decision Tree | Specificity, Precision, Recall, Accuracy 86% (for best classifier Naïve Bayes) |
| [75] | Deanship of E-Learning and Distance Education dataset | 6 | 241 | Mobile Course Access Data, Course Access Data and Assessment Data | decision tree, random forest, sequential minimal optimization, multilayer perceptron, and logistic regression. | TPR, Kappa, Precision, Recall, F1-score, RMSE, Accuracy 99.17 (for best classifier Random Forest) |
| [78] | Learn Moodle Dataset | One | 6119 | Quiz, Forum, Workshop, Assignment, Activeness | Random Forest (RF) and Artificial Immune Recognition System (AIRS), fuzzy unordered rule induction algorithm (FURIA) | Accuracy 99.85% (for best classifier FURIA) |
| Our Study | University of Plovdiv Dataset | One | 7057 records | Attendance, Lectures, Assignments, Code, Exercises, Grade | Random Forest, XGBoost, KNN, and SVM | Precision, Recall, F1-score, Accuracy 78%(for best classifier Random Forest) |

## 6. Conclusions

Early prediction of students' academic performance may help the teachers determine the students who are likely to show poor performance in the final examination. The teacher can then intervene and take necessary measures by paying extra attention to those students. Timely intervention by the teacher can significantly reduce the number of failed students. It is well-known that different machine learning algorithms perform differently with different datasets.

In this work, we have used some statistical and machine learning techniques to analyze and predict students' final grades from the data extracted from logs of Moodle Learning Management System and Zoom reports.

Although the conclusions presented in this paper are for a specific dataset, the study presents a classification technique that can be extended to other LMS and video-conferencing tool logs datasets as our future endeavor.

The study has several limitations. One of the drawbacks lies with our dataset. The size of the dataset is small. The duration of the dataset is only two years. Hence, we will require a larger dataset to carry out some add-on analysis. Our study is a retrospective one. Therefore, early interventions of the students and the remedial classes by the educators were not possible. We shall encourage the educators to apply our model after a specific period so that early intervention is possible. It will help educators to monitor the

performance of the students in a systematic way. The outcome of the study from the point of view of the evaluation metrics is moderate.

In future work, we shall try to collect more data to apply deep learning models to the dataset to enhance prediction accuracy. More features will be collected to find the most significant features that affect the final grades of the learners in the future. In addition, a script that intersects data from Moodle and Zoom in real time settings will be devised.

The presented methodologies also can be extended to other LMS and video-conferencing tool logs datasets with minor feature selection changes.

**Author Contributions:** Conceptualization, S.G., R.D. and S.H.; methodology, A.T., Y.K.S. and S.H.; software, A.T. and N.G.; validation, A.T., S.H. and N.G.; formal analysis, S.G.; investigation, A.T.; resources, A.T.; data curation, P.T., Y.K.S.; writing—original draft preparation, S.G., A.T., S.H., P.T., Y.K.S.; writing—review and editing, S.G., N.G., P.T., Y.K.S. and R.D.; visualization, A.T. and S.H.; supervision, S.G. and R.D.; project administration, S.G., R.D.; funding acquisition, S.G., R.D. All authors have read and agreed to the published version of the manuscript.

**Funding:** The paper is partly supported by the project MU21-FTF-018 "Application of big data analysis methods in higher education" of the Scientific Research Fund at the University of Plovdiv "Paisii Hilendarski".

**Data Availability Statement:** Not applicable.

**Conflicts of Interest:** The authors declare no conflict of interest.

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
