# Peer review of "Exploring Online Activities to Predict the Final Grade of Student"

_mathematics, doi:10.3390/math10203758_

Round 1

Reviewer 1 Report

Review for manuscript mathematics-1896927:

The target of this manuscript is very important for the educational procedure. The subject of predicting the final grades of students is crucial for the teachers, the university decision makers, as well as the students themselves. Using LMS data and Machine Learning methods is a common way to prediction. The combination of these data with recordings concerning the attendance of online lectures can improve the performance of the prediction.

Obviously, the goal of the manuscript is not the presentation of new Machine Learning methods or ensembles but the presentation of a case study based on the dataset obtained from Moodle and Zoom records concerning a university course. These records are well organized while the techniques applied are the usual.

However, the presentation of the manuscript can be improved.

1.       The size of the training set should be defined.

2.       In the text of the manuscript, some figures are cited as "Fig. X", while some of them are referred by using "Figure X". Also, some lectures are named as "Lecture X" while other are named as "lecture X". Assignments are called “assignment X”. I believe that utilizing “Assignment X” is better. In addition, F1-score is named either as "F1-score" or "f1-score".

3.       The tables of data presented as figures contain a "User full name" in each of their entries. Is this "name" an official student's id (which can lead to the identification of the student) or it has been "anonymized" somehow according to European Union’s General Data Protection Regulation?

4.       In the Results section: In the Related Work section, the authors report that cited papers mention that “explainability is the key to make the domain 112 experts understand the reasons behind such predictions”. This is absolutely true. After this, most readers would expect that the authors will try to cope with this task too. Something is written at the end of Univariate part of the Results section, but it is incomplete. Why the submission of Assignment 2 is so important? What about the rest of the assignments?

5.       In the Conclusion section: Concerning RQ3: A comment about the comparison of F1-scores for the methods should be added.

Some more comments and suggestions are the following:

Page 1 - line 37. The acronym "HEIs" has already explained in the Abstract section.

Page 2 - lines 57-58. It is written there: "are Educational Data Mining (EDM) method." The authors probably mean "is Educational Data Mining (EDM) methods".

Page 2 - lines 67-68. It is written there: "and therefore..." This secondary sentence is incomplete.

Page 4 - line 119. What is the meaning of the expression "different machine learning tools"? Different from what?

Page 10 - line 233. It is written there: "The Exercises 5 and Exercises 8". I suggest writing "Exercises 5 and 8"

Author Response

Kindly find our responses in the word file.

Reviewer 2 Report

This study uses statistical and machine learning techniques to predict the students' final grades based on their activities in LMS Moodle and attendance in online lectures. The Chi-square test is utilized to assess the association between students' final grades and event context and the relationships between attendance at lectures and final results. Machine learning algorithms are applied to predict the students' final grades.

Authorst state that the results show correlations between the students' final grades and their activity in the online course and between students' final grades and attendance at lectures. They also state that Random Forest algorithm obtained 70% accuracy.

Authors consider that the predictions are significant for teachers and decision-makers and allow them to take measures to reduce the number of failed students and identify which types of learning resources or student activities are better predictors of the student's academic performance. The study authors also state that the presented methodologies can be extended to other LMS and video-conferencing tool logs datasets with minor feature selection changes.

It is an interesting study. The necessary background is explained. The methodology is adequate to such a study. The results are coherently explained, and adequate for the formulated research questions. The conclusions are coherent to the obtained results.

There are some fails in terms of English grammar that make difficult to understand the contents of the document. Once the text is reviewed, in my opinion, it can be published.

Author Response

Kindly find our responses in the attached word file.

Reviewer 3 Report

The paper describes the results of applying machine learning algorithms to predict student success in the context of a Programming discipline. The authors collected data from Moodle logs of 105 students intersected with their attendances on Zoom online classes. The paper explores the associations between the collected attributes and the final grades of the students, and then describes the results of four machine learning models (Random Forest, XGB, KNN, and SVM). The best model achieved a f-measure of 71%. Among the conclusions the authors mention that students grades is associated with specific types of lectures, exercises and assignments. Moreover, the grades are associated with the attendances of the students.

The work falls under the Educational Data Mining scope and the many studies of the Learning Analytics field that seek to early detect students at-risk of fail.

I believe the paper does not fit for publication. The main reason for that is that some of the answers for the research questions can not be derived from the experiments presented in the paper.

To develop the models reported here, the authors used information from the students that covers the whole semester of the discipline. Considering that the main goal of such models is to early detect students at-risk, it is key to evaluate how the models are able to detect such students in the very first weeks of the semester/year. It is not clear in the paper, how the current models can effectively help on that direction. It would be important the authors present the performance of the models progressively for each period of the discipline (week1, week2, etc), so that it would be possible to see how the models can identify the students at-risk in the very beginning of the course. A number of papers published in several journal have already covered this issue (including papers published here in MDPI, Applied Sciences and Information journals). In my opinion, the answers to the research questions RQ3 and RQ4 do not reflect the results presented in the paper.

The most interesting aspect of the paper is the intersection of data coming from Moodle and Zoom. The paper does not clarify how that intersection was made, but I assume they developed a script to do that. How such integration could be made in real-time settings? Is it possible to perform such integration of logs of Moodle and Zoom so that the models would receive such information in time of perform the predictions? I believe that this is possible, but the paper do not discuss any of that challenges and answers on RQ5 that this strategy can be even extended to other LMS and video-conferencing tools.

The paper does not present any discussion of the findings, or any comparison of the results with the previous literature and jumps from reporting the results in section 4 directly to the conclusions. The most important findings of the paper that are presented in RQ1 and RQ2 are largely covered in the literature. The authors should have explored that in depth, but did not tackle that at all.

Other comments:

In the related literature, the authors must observe whether the results they mention used the appropriate metrics to evaluate the performance of the models. It is a common mistake in may papers to report the general accuracy of models even if they were trained with unbalanced data.

In the future, the authors could provide a table contrasting the different results achieved by the many works mentioned in the related literature, together with the differences and similarities regarding the contexts in which they were developed (size of the database, educational context, discipline, number of students, performance metrics, algorithms used to generate the models, most important attributes used as input in the models, data source, among others).

English

logs for their activities -> logs of their activities

if the student is read each

Author Response

The responses are attached as a word document. Kindly look at it.

Reviewer 4 Report

The paper describes an exciting work, which contributes to the improvement of the actionability of decision-makers and teachers in HEIs. Concretely, the aim is to predict students' final grades based on their activities in both Moodle (LMS) and Zoom sessions. Results provide evidence regarding a positive relationship between students' behavior and academic performance. As well as, results give evidence of a good prediction power of machine learning models, which are considered. 

However, there are some issues that are not properly addressed. Firstly, the introduction is too general (decisions and actionability) regarding the main scope of the paper (final grade prediction). Secondly, the research questions must be improved and specified, because they are too open. Thirdly, the gap of knowledge is not identified regarding the related work. Fourthly, the instructional design of the course has a lot of impact on students' behavior, but it is not described at all. Fifthly, there is no ethical consent by participants. Sixthly, there is low replicability, because the dataset is not properly described, statistical methods are not properly used and the machine learning algorithms setup is not completely described (eg. training size, validation method, etc). Finally, there are no discussions, limitations, or further works.

More specifically, 

1. A more concrete definition of research questions is required. The RQ1 is not properly defined and not at all answered, the relation is between students' behavior against learning resources and final grades. Why the features are dichotomic? Even the attendance. RQ2 and RQ3 have been answered in several works. What are the differences here? RQ4 is not answered. There is no analysis of the early predictions. RQ5 is not answered and is too general. 

2. A more detailed description of feature engineering in related work is required. Why are considered such features? 

3. The dataset is not balanced. How are you managing such an issue?

4. The machine learning setup is not properly defined. Training size is not defined, validation method is not described, and results by class are not described (which class is better?).

5. The conclusions must be improved.

Author Response

(The authors gave the same response as above.)

Round 2

Reviewer 3 Report

The work has improved significantly and many of the considerations made in the previous review were tackled in this new version.

The methodology is still confusing. The authors mention they split the data into train and test but also used 5-fold cross-validation. This is a bit unusual, as the k-fold cross-validation approach already splits the dataset into k-folds. Did they applied 5-fold after splitting the dataset into train and test? In case so, why? It is necessary the authors explain their methodology regarding this aspect.

Moreover, there is a huge number of attributes considered as input for the models. Did the authors apply feature selection algorithms to reduce the dimensionality of the dataset? In case not, wasn’t that the case for that? The dimensionality of the data influences in the performance of the models.

There is no need of a pie chart to say the data was balanced.

How many weeks has the course? To which point the results found are really useful considering the goals of the study?

Author Response

Kindly look at the attached word file.

Reviewer 4 Report

Almost every comment was appropiately addressed. The work is well defined and limitations are properly identified. It is required be careful with ethical issues in studies with humans, for instance, informed consent by participant.

Nevertheless, novelty, generalization and reproductibility are not clear. To reproduce, it is not required the algorithm definition, they are well known. It is required a more detailed definition of training and evaluation method of machine learning algorithms (methodology section). For instance, training data size, instance selection and validation strategy (cross). 

Can you provide the dataset?

Author Response

Kindly look at the attached word file.

Round 3

Reviewer 3 Report

The authors have attended the main considerations of my previous reviews. 

Reviewer 4 Report

The paper provides a well described analysis the behavior of students. Also, it describes the application of learning algorithms to predict the grades based on such behavior. 

Cross validation doesn't improve the performance of the algorithms, at all. This is a hugh mistake.

Author Response

(The authors gave the same response as above.)
